# Dealing with Label Scarcity in Computational Pathology: A Use Case in Prostate Cancer Classification

**Koen Dercksen**                                    KOEN.DERCKSEN@RADBOUDUMC.NL
**Wouter Bulten**                                    WOUTER.BULTEN@RADBOUDUMC.NL
**Geert Litjens**                                    GEERT.LITJENS@RADBOUDUMC.NL

## Abstract

Large amounts of unlabelled data are commonplace for many applications in computational pathology, whereas labelled data is often expensive, both in time and cost, to acquire. We investigate the performance of unsupervised and supervised deep learning methods when few labelled data are available. Three methods are compared: clustering autoencoder latent vectors (unsupervised), a single layer classifier combined with a pre-trained autoencoder (semi-supervised), and a supervised CNN. We apply these methods on hematoxylin and eosin (H&E) stained prostatectomy images to classify tumour versus non-tumour tissue. Results show that semi-/unsupervised methods have an advantage over supervised learning when few labels are available. Additionally, we show that incorporating immunohistochemistry (IHC) stained data provides an increase in performance over only using H&E.

**Keywords:** computational pathology, prostate cancer, unsupervised learning, deep learning

## 1. Introduction

Prostate cancer is manually graded by pathologists on H&E stained specimens, based on the morphological features of epithelial tissue. Since this is a labour intensive process, an automated system to perform cancer grading would be of great value. However, to develop such systems typically large sets of labelled data are required. To collect these data, annotations from human experts (in this case uropathologists) would be required. Such expertise is rare, and thus creating the required labelled datasets is challenging. This inherently limits the potential for algorithm development. (Litjens et al., 2017).

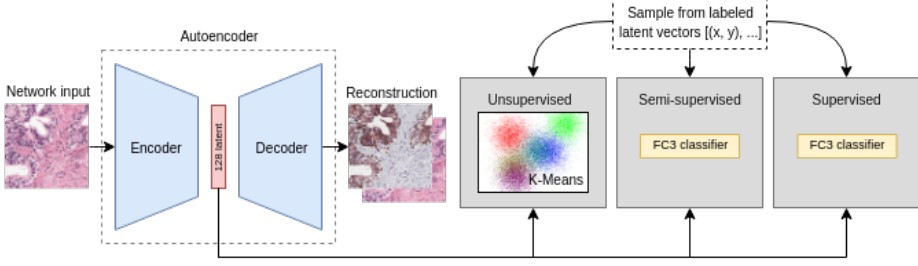

Figure 1: The flow of data for each of the three methods. Note that the data flow for the supervised method is identical to that of the semi-supervised method, but does not utilise unsupervised pre-training.

We hypothesise that a semi- or unsupervised approach, leveraging unlabelled data, can learn a good latent tissue representation which can be used to classify unseen tissue, without using labelled data during training (Arevalo et al., 2015; Hou et al., 2019; Kallenberg et al., 2016). While a supervised approach undoubtedly outperforms unsupervised methods given enough data, we show the advantage of using a semi- or unsupervised approach when little labelled data is available through a three-way comparison (Figure 1). In addition to H&E, we test incorporating IHC stained images that highlight epithelial tissue in order to force learning a more descriptive latent space.

## 2. Methodology

**Data.** We used the PESO dataset (Bulten et al., 2019), which consists of 102 registered whole-slide image (WSI) pairs from patients that underwent a radical prostatectomy. Each pair is made up of a H&E slide, and a slide that was processed using IHC with an epithelial (CK8/18) and basal cell (P63) marker. The set was divided into 62 training and 40 test slides. All H&E slides are publicly available.[1]

For training, two separate datasets were created: one completely randomly sampled set $D_r$ of 100.000 patches without labels. This set is used to pre-train the semi- and unsupervised methods. Another labelled set $D_b$ of equal size was created in which the ratio of stroma, benign epithelium and tumour tissue (class label determined by the center pixel) is $\{0.25, 0.25, 0.50\}$ respectively. Subsets of varying sizes were used for the experiments in this paper. All patches have a size of $256 \times 256$ pixels (pixel resolution 0.96 $\mu$m) and are sampled pair-wise from both stains.

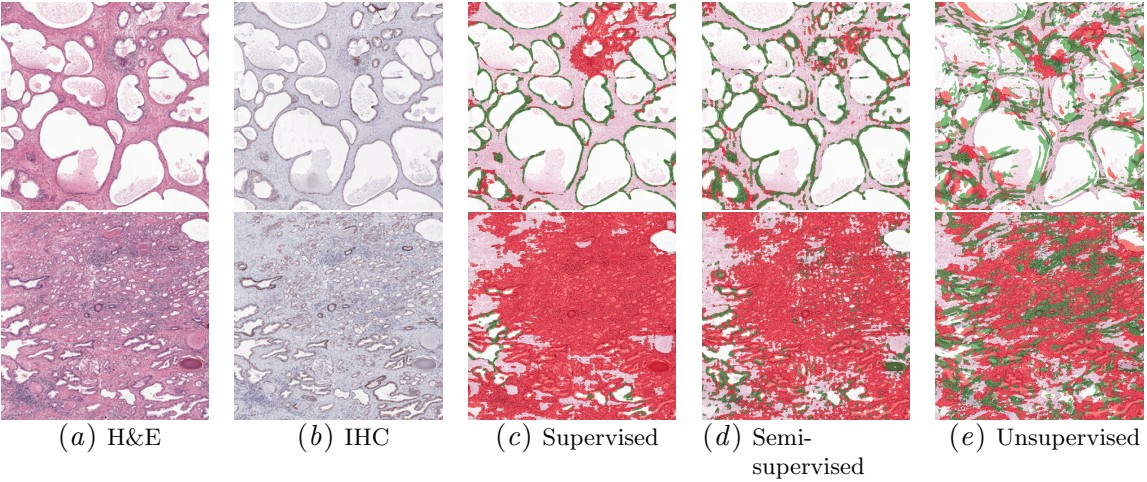

$(a)$ H&E  $(b)$ IHC  $(c)$ Supervised  $(d)$ Semi-supervised  $(e)$ Unsupervised

Figure 2: Classification maps of models trained with 1000 labelled patches applied to a benign (top row) and tumour (bottom row) region (transparent = stroma, green = benign epithelium, red = tumour).

**Semi- & unsupervised training.** An autoencoder $M$ is trained on $D_r$ to reconstruct either H&E or IHC patches given an H&E input patch by optimising the mean-squared error (MSE). The encoder part of the network $M_e$ consists of strided convolution layers to

---

1. https://doi.org/10.5281/zenodo.1485967

Table 1: F1 scores for all methods trained using various subsets of $D_b$. NLP = Number of labelled patches, SV = supervised.

| NLP | H&E → H&E | | H&E → IHC | | H&E | IHC |
|---|---|---|---|---|---|---|
| | Semi-SV | Un-SV | Semi-SV | Un-SV | SV | SV |
| 100 | $0.56 \pm 0.07$ | $\mathbf{0.71 \pm 0.02}$ | $0.69 \pm 0.04$ | $\mathbf{0.71 \pm 0.02}$ | $0.00 \pm 0.00$ | $0.00 \pm 0.00$ |
| 500 | $0.70 \pm 0.03$ | $0.72 \pm 0.02$ | $\mathbf{0.75 \pm 0.01}$ | $0.74 \pm 0.01$ | $0.58 \pm 0.12$ | $0.67 \pm 0.09$ |
| 1000 | $0.73 \pm 0.01$ | $0.72 \pm 0.01$ | $\mathbf{0.77 \pm 0.01}$ | $0.74 \pm 0.01$ | $0.76 \pm 0.01$ | $\mathbf{0.77 \pm 0.02}$ |
| 2000 | $0.74 \pm 0.01$ | $0.74 \pm 0.02$ | $\mathbf{0.77 \pm 0.01}$ | $0.75 \pm 0.00$ | $0.73 \pm 0.03$ | $0.56 \pm 0.27$ |
| 10.000 | $0.76 \pm 0.01$ | $0.70 \pm 0.01$ | $\mathbf{0.78 \pm 0.01}$ | $0.74 \pm 0.01$ | $0.74 \pm 0.04$ | $0.71 \pm 0.29$ |
| 100.000 | $0.75 \pm 0.02$ | $0.73 \pm 0.01$ | $0.74 \pm 0.02$ | $0.75 \pm 0.01$ | $0.88 \pm 0.02$ | $\mathbf{0.91 \pm 0.02}$ |

compress the input into the 128-dimensional latent space. The decoder contains convolution and upsampling layers to decompress the latent vector back to the original input size. After training, the latent vectors $M_e(D_r)$ are clustered using k-means with 50 clusters. Finally, the clusters are assigned labels through majority voting by using subsets of varying sizes from $D_b$. Empty clusters are assigned the *stroma* label. For the semi-supervised experiments, the same autoencoder $M$ is trained, but instead of using k-means labels are now assigned by training a single-layer classifier on subsets of $M_e(D_b)$.

**Supervised training.** As a baseline, only $M_e$ is used and trained in a supervised fashion on subsets of $D_b$ end-to-end, without using unsupervised pre-training on $D_r$. This acts as an upper-bound to the classification performance on this dataset.
Every experiment uses data augmentation (flipping/hue/saturation/brightness/contrast) and is repeated five times in order to report confidence intervals.

**Validation.** We sample 10.000 patches from the PESO test regions with the same tissue ratio as $D_b$ to measure the final performance of each approach. Every method is trained to predict all three classes (aiming to learn the difference between benign epithelium and tumour), and the F1 score is reported for tumour versus non-tumour classification. At test-time, all models except for the supervised IHC network are validated on H&E.

## 3. Results & Discussion

Semi- and unsupervised methods have an advantage over supervised training when few labels are available (Table 1). The semi-supervised method reaches an F1 score of 0.75 with as few as 500 labelled patches, compared to 0.58 for the supervised H&E classifier.
Additionally, the semi-/unsupervised performance is more robust than the supervised approach, which became unstable at small dataset sizes. At large dataset sizes the supervised method performed substantially better than the other approaches, as expected.
Using IHC data as a reconstruction target (or input for the supervised approach) improves the performance of every method, indicating that the extra information present in the IHC data leads to better latent representations.
At larger labelled dataset sizes the performance of the semi- and unsupervised approaches seems to saturate. This can be caused by the low complexity of the classification models (k-means and single layer neural network) or by limitations in the representative power of the latent space. In future work it might be interesting to investigate the latent representations across the different methods to better understand this phenomenon.

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
