# OpenReview forum: "Dealing with Label Scarcity in Computational Pathology: A Use Case in Prostate Cancer Classification"
_MIDL.io/2019/Conference/Abstract — MIDL Abstract 2019_

### Official Review · AnonReviewer2 · 2019-04-27
**Nice work for improving data efficiency**

**Rating:** 4
**Confidence:** 3

**Review:**

The paper proposes to use AE for feature learning and compare three different ways (unsupervised, semi and fully supervised) for incorporating the features for segmentation. It is an interesting exploration for small data learning.

---

### Official Review · AnonReviewer1 · 2019-05-01
**yet another evidence showing advantages of semi-supervised/unsupervised learning**

**Rating:** 3
**Confidence:** 2

**Review:**

This paper investigates the performance of unsupervised and supervised deep learning methods when labeled data is scarce.  Authors compared three methods with an application of prostate cancer (with histopathology images):
1) clustering autoencoder latent vectors (unsupervised),
2) a single layer classifier combined with a pre-trained autoencoder (semi-supervised), and
3) a supervised CNN.

At the end of experiments/comparisons, authors showed that when there is few labels, generalization ability of unsupervised and semi-supervised methods are better than supervised method. This is known fact with a few publications already in medical imaging field, with several others in vision. However, it maybe a new finding for histopathology image analysis in particular, therefore the paper is of significant in that sense.

Given the page limit, experiments are convincing, and enough details are given for readers to understand what is going on.

---

### Decision · Program_Chairs · 2019-05-06
**Acceptance Decision**

Accept